# Health status outcome among cannabis addicts after treatment of addiction

**Mohammed Ali A. L. Siwar**[1,2◉], **Sayed Hossam El-Din Mansour**[3], **Maha A. Aldubayan**[1], **Ahmad H. Alhowail**[1], **Yasser S. Almogbel**[4], **Ashraf Mahmoud Emara**[1,5◉] *

**1** Department of Pharmacology and Toxicology, College of Pharmacy, Qassim University, Buraydah, Qassim, Saudi Arabia, **2** Asir Central Hospital, Asir, Saudi Arabia, **3** Psychiatry Department, Al Amal Hospital for Mental Health, Asir Region, Saudi Arabia, **4** Department of Pharmacy Practice, College of Pharmacy, Qassim University, Buraydah, Saudi Arabia, **5** Department of Forensic Medicine and Clinical Toxicology, Faculty of Medicine, Tanta University, Tanta, Egypt

◉ These authors contributed equally to this work.
* a.omara@qu.edu.sa

**Data Availability Statement:** All relevant data are within the manuscript and its Supporting information files.

## Abstract

The abuse of Cannabis is a widespread issue in the Asir region. It has a lot of legal and occupational repercussions. The purpose of this study was to evaluate the health status of cannabis addicts at admission and after treatment using body mass index, glycemic status, liver function, renal function, and oxidative stress. A cross-sectional study was conducted with 120 participants. The study was conducted at Al Amal Hospital for Mental Health in Asir region of Saudi Arabia, with 100 hospitalized patients receiving addiction treatment and 20 healthy volunteers. The participants were divided into two groups: group I, the control group, and group II, the cannabis addicts. The socio-demographic data were gathered. The level of cannabis in the urine and the CWAS [Cannabis Withdrawal Assessment Scale] were determined. In addition, the Body Mass Index [BMI], vital signs [temperature, heart rate, systolic blood pressure, diastolic blood pressure, and respiratory rate], serum levels of albumin, total bilirubin, direct bilirubin, AST, ALT, and ALP, urea, creatinine, Thiobarbituric acid-reactive substances [TBARS], superoxide dismutase [SOD], reduced glutathione [GSH], and catalase [CAT] were analyzed on the first day of admission and after treatment. According to the results, there was no significant change in the body mass index. The vital signs in the cannabis user group were significantly lower than the corresponding admission values. Regarding renal function tests such as urea and creatinine, we found that after treatment, the mean urea and creatinine values in the cannabis user group did not differ significantly from the corresponding admission values. However, after treatment, the mean values of fasting blood glucose levels in the cannabis user group were significantly lower than at admission. Also, the mean values of liver function tests such as albumin, total bilirubin, direct bilirubin, AST, ALT, and ALP in the cannabis user group were significantly lower than the corresponding admission values after treatment. In assessing the antioxidant system, we found that the mean values of TBARS, SOD, GSH, and CAT in the cannabis user group did not differ significantly from the corresponding admission values after treatment. The current findings have revealed that cannabis addiction harms the various body systems and has significant

**Funding:** The researchers would like to thank the Deanship of Scientific Research, Qassim University for funding publication of this project.

**Competing interests:** The authors have declared that no competing interests exist.

implications for the addict's state of health. The values of oxidative stress biomarkers did not change in this study, but other measured parameters improved after treatment.

## Introduction

Generally, the terms "marijuana" and "cannabis" refer to smoking-like preparations of the leaves and flowers of Cannabis sativa, in which the active component is 9-tetrahydrocannabinol [9-THC], which is known to cause intoxication. The effects of cannabis toxicity depend on numerous variables, including dosage, method of administration, a user's previous use of the drug, expectations, attitudes toward the drug, and the social environment when using it [1].

Cannabis is commonly inhaled as smoke due to the rapid action of this mode of administration. It can also be taken orally as "grass" brownies or tea. Cannabis use has both recreational and therapeutic purposes. However, its use is always associated with acute or chronic toxic effects [2].

Cannabis is the most commonly used illicit substance worldwide. Approximately 147 million people [2.5% of the global population] use it. Men are more prevalent users than women, and it is commonly used by both adolescents and adults [3]. Recently, the use of both amphetamines and cannabis has increased in Saudi Arabia [4, 5].

According to a 2018 retrospective study conducted by the Buraydah Psychiatric Rehabilitation Center [PRC] in Saudi Arabia, amphetamine, alcohol, and Cannabis are the most commonly abused substances [4]. About 75% of abusers were 20–40 years old; 66% had dropped out of high school, 16% from elementary school, and 9.6% from college [6].

A recent meta-analysis concluded that acute 9-THC exposure causes tachycardia in humans. The cardiovascular effects of 9-THC are typically dose-dependent [7]. Exposure to Cannabis raises the heart rate and blood pressure and can sometimes cause orthostatic hypotension. Cannabis users have reported severe cardiac problems, including acute coronary syndromes and strokes [8].

In vitro studies, however, show that it decreases the viability of, induces oxidative stress in, and suppresses apoptosis and mitochondrial function in human epithelial lung cells [9]. 9-THC also induces in vitro CYP450 1A1, which activates polycyclic aromatic hydrocarbons and is associated with lung cancer [10].

In young populations, synthetic cannabinoid toxicity should be considered a significant cause of acute kidney injury (AKI). Several cases of AKI have been reported in association with the use of synthetic cannabinoids (SC), which appear to be more dangerous than natural Cannabis [11]. While long-term effects remain unknown, SC-associated AKI is potentially reversible with supportive treatment [12].

9-THC and endogenous cannabinoids stimulate the endogenous cannabinoid receptor system to increase appetite and promote adipogenesis and body fat production [13]. Obesity, particularly central adiposity, is the leading risk factor for developing type 2 diabetes. Due to the known effects of cannabinoids on adipose tissue and glucose-insulin metabolism, Sidney [2016] established a plausible association between Cannabis use and diabetes [14].

The chronic effects of Cannabis have been evaluated in various studies. Cannabis addiction has toxic effects on many organs, including the brain, heart, liver, and kidneys. Also, it induced an alteration in body mass index, redox balance, and glycemic status. However, there are no studies in the literature on the health status outcomes of Cannabis addicts after addiction treatment. The chronic toxic effects of Cannabis addiction on different body systems may not be adequately managed after standard addiction treatment. By evaluating body mass index,

glycemic status, liver function, renal function, and oxidative stress, this study aims to determine the effect of Cannabis treatment on the health status of Cannabis addicts.

## Subjects and methods

### Subject

This cross-sectional study was conducted at Al Amal Mental Health Hospital, Asir region, KSA. Twenty healthy male individuals as a control group [group I] and a hundred male cannabis abusers [group II] were included. All participants in this study had similar food habits and were devoid of any underlying medical conditions. Cannabis abusers were admitted for cannabis addiction treatment. This study was conducted over two years. The medical history of all the participants was recorded. An experienced psychiatrist interviewed the participants and evaluated them using DSM-5 criteria. Every day, the presence of abstinence syndrome was checked. In terms of lifestyle and age, the participants in the cannabis user and control groups were frequency-matched. Each participant's general medical history, smoking status, and substance use history were gathered (Fig 1). The participants who have the following are excluded from the present study: given any therapeutic agents that could potentially change biochemical parameters, seriously ill patients [e.g., agitation, convulsion, disturbed consciousness, severe viral infection], and who are less than 18 years old and more than 60 years old.

### Treatment patients received

The treatment for cannabis use disorder includes psychotherapy and pharmacotherapy. It mainly focuses on psychotherapy treatments such as motivational enhancement therapy, cognitive behavioral therapy, and contingency management. Pharmacotherapy has been performed as an adjunctive intervention to psychosocial treatment. Pharmacotherapy focuses on treating withdrawal symptoms such as irritability, anxiety, restlessness, appetite changes, and

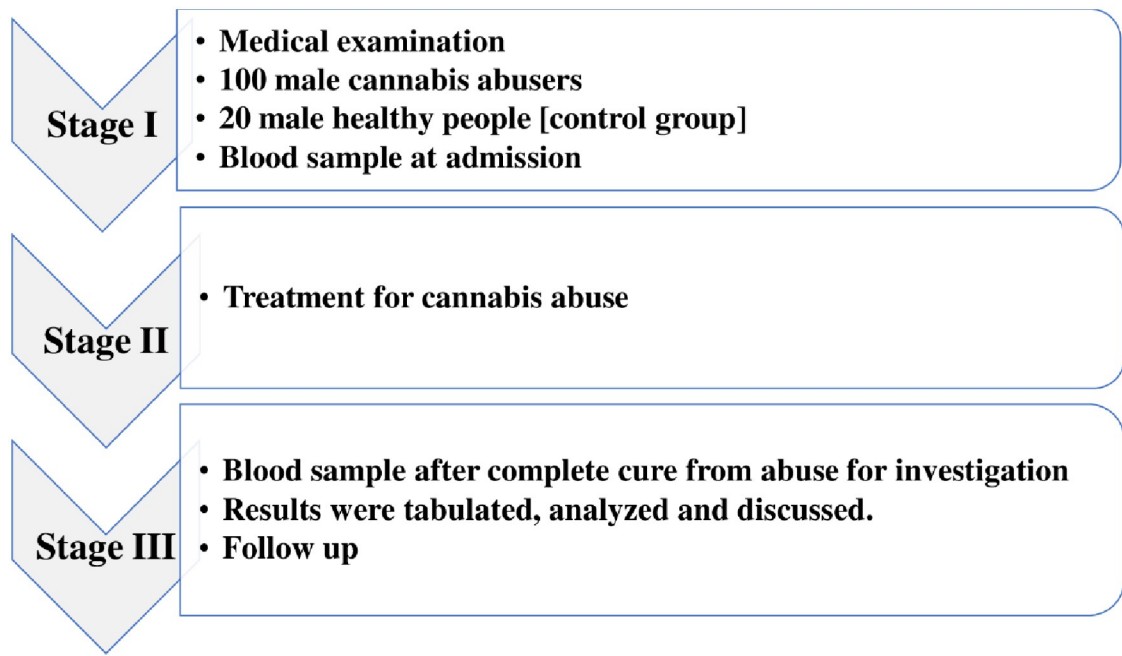

**Fig 1. Schematic diagram of the study design.**

disrupted sleep. It includes Baclofen, Zolpidem, Lofexidine, and Quetiapine. The follow-up for patients was once per week.

## Medical examination

The history of the participants was recorded from their medical records, including age, sex, individual habits, occupation, toxicological history, medical disease, and drug treatments. In addition, a complete physical examination was recorded. Finally, BMI [kg/m2] was measured using the participant's height and weight.

## Blood sample

Blood samples were collected using sterile, disposable syringes and clean venepuncture under aseptic conditions. Each participant and the control sample had roughly 10 ml of blood drawn from them. Blood was placed in clean, dry, silicon-coated test tubes and allowed to clot for 20 minutes at room temperature. Next, the blood was centrifuged for 10 minutes at 2000 rpm, and the supernatant was aliquoted into 1.7-ml Eppendorf tubes. The serum samples were then placed in tightly sealed vials and stored at -80˚C until analysis. The samples were collected both at the time of admission and after treatment.

## Urine sample

Each urine sample [20 mL] was collected in a polypropylene container and refrigerated at -20˚C immediately following the urination until analysis. The samples were taken at the time of admission.

## Determination of urinary 9-THC

The THC concentration was measured using a Randox biochip array. Following the kit instructions, nine calibrators included with the Randox kits were prepared and analyzed with each batch of samples. Then, phosphate-buffered saline and 0.1% Triton X-100 [2 mL] were added to each tube. After 30 seconds of vortex mixing. Then, the tubes were centrifuged at 14,000 rpm at 0˚C for 15 minutes. Polypropylene tubes measuring 5 mL were filled with the recovered supernatant. Individual Randox DOA I biochips containing antibodies against 9-carboxy-11-nor-9-THC were pipetted with 120 μL of Randox assay buffer and 60 μL of urine. Each chip received 120 microliters of the conjugate. The entire rack of chips [6 9-chip carriers containing nine calibrators and 48 extracted fortified samples or specimens, a total of 54 samples] was placed on the thermoshaker for 30 minutes at 330 rpm and 25 degrees Celsius. The chip carrier was extracted from the thermoshaker and washed with Randox wash buffer six times. Next, the signal reagent was mixed according to the kit's instructions, and nine chips received 250 μL. [one carrier at a time]. Each carrier was incubated for two minutes in the dark before being placed in the Evidence Investigator for signal analysis. The results were determined by comparing the signal from the sample chips to the signal produced by the cali-brators [15–17].

## Liver function tests

The serum albumin concentration was determined colorimetrically using a commercial kit supplied by Diamond [Egypt] [18]. The total bilirubin and direct bilirubin concentrations were determined colorimetrically using kits obtained from Bio-diagnostic, Egypt [19]. A kit supplied by Randox [Egypt] was deployed to estimate the serum activity of AST and ALT col-orimetrically [20]. The serum level of ALP was determined using a commercially available kit

[BioMérieux Co., France] and adopting the method used by Belfield and Goldberg [1971] [21]. Using a Randox kit, the serum was analyzed to determine the blood glucose level colorimetrically [22].

## Kidney function tests

The serum concentration of urea was measured using a commercially available kit supplied by Diamond [Saudi Arabia] [23]. The serum creatinine concentration was determined using a commercially available kit [Diamond, Saudi Arabia] [24].

## Determination of enzymatic and non-enzymatic antioxidants

**Determination of serum thiobarbituric acid reactive substance.** According to the method of Esterbauer and Cheeseman, the human thiobarbituric acid reactive substance [TBARS] ELISA kit was used for quantitative measurement [25]. The thiobarbituric acid reactive substance [TBARS] ELISA kit was purchased from MyBiosource, Inc. [CA, United States of America] [catalog number MBS166987]. Bringing all reagents to room temperature was the first step in the preparation process. The experiment was then conducted at this temperature. Creating a 16 nmol/m1 standard stock solution, 120 µL of the standard [32 nmol/ml] was reconstituted with 120 µl of the standard diluent. Before preparing the dilutions, the standard was allowed to rest for 15 minutes while being gently agitated. Then, duplicate standard points were prepared by serially diluting the standard stock solution [16 nmol/m1] 1:2 with standard diluent to yield solutions with 8 nmol/ml, 4 nmol/ml, 2 nmol/ml, and 1 nmol/ml. The zero standard was standard diluent [0 ng/ml]. Afterward, the concentrated cleaning solution was diluted with double-distilled water [1:25]. Next, 50 µl of standard and 40 µl of sample-to-sample were added to the standard well and sample-to-sample well, respectively. Next, 10 µl of anti-TBARS antibodies were added to the sample, followed by 50 µl of streptavidin-HRP and thorough mixing. The plates were sealed and incubated for sixty minutes at 37˚C. The plates were then washed five times for one minute each with 0.35 ml of washing buffer. After that, 50 µl of substance solution A and 50 µl of substance solution B were added to each well, respectively. The plate was sealed and incubated in the dark at 37˚C for 10 minutes. Each well was then filled with 50 µl of the stop solution. Within 10 minutes of adding the stop solution, the optical density [OD value] of each well was determined using a microplate reader calibrated to 450 nm.

**Determination of serum reduced glutathione.** Quantitative measurement was performed with the human reduced glutathione [GSH] ELISA kit. This item was purchased from MyBiosource, Inc. [California, United States of America] [catalog number MBS727656]. All reagents were brought to room temperature before preparation. The experiment was then performed at this temperature. The required number of coated wells in the holder was secured, and 100 µL of standard or sample solution was added to each well. The standard bottle was manually shaken gently and pipetted up and down the solution three times before being added. 100 µL of phosphate-buffered saline [pH 7.0–7.2] was added to the blank control well. 50 µL of the conjugate was added to the respective sample wells and mixed thoroughly. The plate was covered and incubated at 37˚C for one hour. Within 10 minutes of the addition of the stop solution, the optical density [OD value] of each well was measured using a microplate reader [set to 450 nm].

**Determination of serum superoxide dismutase.** The human superoxide dismutase [SOD] ELISA kit was used for this quantitative measurement, according to Misra and Fridovich [26]. This was sourced from MyBiosource, Inc. [CA, United States of America] [catalog number MBS2707322]. All reagents were brought to room temperature before the experiment,

and the test was then conducted at this temperature. Each well received 100 μL of the standard or sample and was incubated for one hour at 37˚C. Then, 100 μL of detection reagent A was aspirated and added. The mixture was incubated at 37˚C for one hour before being washed three times. 100 μL of the prepared detection reagent B was added, incubated at 37˚C for thirty minutes, and washed five times. At 37˚C, 90 μL of substrate solution was added and incubated for 10 to 20 minutes. Within 10 minutes of adding 50 μL of stop solution to each well, the optical density [OD value] was determined using a microplate reader set to 450 nm.

**Determination of serum catalase.** According to Aebi, the human catalase [CAT] ELISA kit was used for this quantitative measurement. It was obtained from MyBiosource, Inc. [CA, United States of America] [catalog number MBS165657]. During preparation, agents were brought to room temperature. The standard well was then supplied with 50 μL of standard, in addition to adding a 40 μL sample to the sample wells, 10 μL of anti-CAT antibody was also added. Then, 50 μL of streptavidin-HRP was added to each of the standard and sample wells. The resulting mixture was then thoroughly combined. The plate was sealed and incubated at 37˚C for 60 minutes, after which the sealer was removed and the plate was washed five times with a wash buffer. The wells were soaked for 1 minute per wash in 0.35 ml of wash buffer [for a total of five times]. The plate was then blotted with paper towels, and 50 μL of substrate solution A and 50 μL of substrate solution B were added to each well. Next, the plate was sealed with a new sealant and incubated at 37˚C for 10 minutes in the dark. The stop solution was then added to each well at a volume of 50 μL. Within 10 minutes of adding the stop solution, the optical density [OD value] of each well was determined using a microplate reader calibrated to 450 nm [27].

## Ethics statement

This study was approved by the College of Pharmacy, the ethics committee of Qassim University, and the Research Ethics Committee of the Asir region, Ministry of Health for Saudi Arabia. All evaluations were conducted following the guidelines for human research established by the Ethical Committee of the College of Pharmacy, Qassim University, and The Research Ethics Committee at the Ministry of Health for Saudi Arabia. Consent will be obtained from all participants in this study before their participation. At any time, the patient may be withdrawn from the study. The withdrawn patient will be replaced by another patient after obtaining their consent. All patients will be coded for privacy purposes. The outcomes will be made public, following international policy.

## Statistical analysis

Data were collected, tabulated, and statistically analyzed using the paired samples t-test to determine whether there were any significant differences between the data of the cannabis user group before and after treatment. Results will be expressed as means ± SD, and statistical significance will be determined if the p-value is less than 0.05. In addition, Pearson correlation coefficients were used to determine the relationships between the CWAS, THC levels, age, and duration of addiction with BMI, vital signs, renal, and hepatic parameters. The SPSS Version 21.0 [SPSS, Inc., Chicago] package will be utilized for all statistical analyses.

## Results

### Socio-demographic characteristics

**Age.** In this study, the ages of the participants ranged from 18 to 45 years old, with mean ages of 25.00 ± 7.39 and 25.00 ± 10.95 years in the control group and the cannabis user group,

respectively. Table [1] showed that most patients' ages ranged between 26 and 35 years old, representing 39% of the cannabis user group. There were no significant changes between the control group and the cannabis user group as regards the mean age [p > 0.05].

**Residence.**   Table [1] showed that most studied participants were from urban areas in both the control and cannabis user groups (65 and 79%, respectively). While the rest were from rural areas (35% and 20% in the control and cannabis user groups, respectively).

**Marital status.**   Regarding the distribution of the study participants according to their marital status, the present study showed that the majority of participants were divorced and singles in the cannabis user group, with percentages of 34 and 27, respectively, as shown in Table [1].

**Social status.**   In the present study, most participants lived with their friends (45% and 38% in the control and cannabis user groups, respectively) [Table 1].

**Educational level.**   Regarding the distribution of the study participants according to their educational level, the present study revealed that the highest percentages of participants were graduates and high school students, with percentages of 52 and 47, respectively. In contrast, the lowest percentages were for patients who were non-educated, as demonstrated in Table [1].

**Occupational status.**   In the present study, the studied participants were employed (65% and 52% in the control and cannabis user groups, respectively), while others were unemployed (35% and 48% in the control and cannabis user groups, respectively) [Table 1].

**The duration of cannabis intake.**   In this study, the duration of cannabis intake by the participants ranged from 1 to 5 years, with a mean age of 3.37 ± 1.58 in the cannabis user

**Table 1. Distribution of the control and the cannabis user group s according to their ages, residence, marital status, social status, educational levels, occupational status, and special habits.**

| Items | | Group I [n = 20] | Group II [n = 100] |
|---|---|---|---|
| **Age [years] Mean ± SD** | | 25.00 ± 7.39 | 25.00 ± 10.95 |
| **Residence [%]** | **Urban** | 65 | 79 |
| | **Rural** | 35 | 21 |
| **Marital status [%]** | **Single** | 45 | 27 |
| | **Married** | 50 | 23 |
| | **Divorced** | 0 | 34 |
| | **Separated** | 5 | 16 |
| **Living status [%]** | **Living with family** | 25 | 29 |
| | **Living alone** | 30 | 33 |
| | **Living with friends** | 45 | 38 |
| **Educational levels [%]** | **Non-educated** | 0 | 1 |
| | **High school** | 40 | 47 |
| | **Graduate** | 60 | 52 |
| | **Postgraduate** | Nil | Nil |
| **Occupational status [%]** | **Student** | Nil | Nil |
| | **Employed** | 65 | 52 |
| | **Unemployed** | 35 | 48 |
| **Special habits [%]** | **Smoker** | 100 | 100 |
| | **Caffeine user** | 100 | 100 |

Group I–Control group; Group II–the cannabis user group.

* Significant at p<0.05 level.

**Table 2. Statistical distribution of the the cannabis user group [n = 100] according to the duration of cannabis intake [years].**

| Duration of intake [years] | Group II [n = 100] | | |
|---|---|---|---|
| | No. | % | Mean ± SD |
| ≤ 1 year | 13 | 13 | 3.37 ± 1.58 |
| 2–3 years | 42 | 42 | |
| 4–5 years | 36 | 36 | |
| > 5 years | 9 | 9 | |

Group II–Cannabis group

group. Table [2] showed that the majority of patients with a duration ranging between 2 to 3 and 4 to 5 years represented 42% and 36% respectively.

**Motives for substance intake.** Regarding the motives of the studied patients for intake, lifestyle stress and the influence of friends represented the most common causes for starting and continuing intake (33%, and 30% in the cannabis user group, respectively). In addition, frustration and relationship difficulties were fundamental reasons for substance use disorder, representing 21% and 16% of the cannabis user group, respectively, as shown in Table [3].

**Tetrahydrocannabinol level and Cannabis Withdrawal Assessment Scale.** The present study demonstrated that the mean tetrahydrocannabinol levels were 96.68 ± 26.47 in the cannabis user group at admission. The mean Cannabis Withdrawal Assessment Scale value was 13.43 ± 5.48 in the cannabis user group at admission [Table 4].

**Body mass index [BMI].** As shown in Fig 2 and Table 5, there was no significant change in the mean BMI values at admission and after treatment in the cannabis user group compared with the control group [p = < 0.05]. The mean BMI values after treatment in the cannabis user group demonstrated no significant change compared with the corresponding admission values [p = < 0.05].

## Vital signs

The current study demonstrated that there was no significant change in the mean temperature values at admission and after treatment in the cannabis user group compared with the control group [p = < 0.05]. On the other hand, the mean temperature values after treatment in the cannabis user group demonstrated a significant decrease compared with the corresponding admission values [p = < 0.05] [Fig 2 and Table 5].

As shown in Fig 2 and Table 5, there was a significant increase in the mean heart rate values in the cannabis user group at admission and after treatment compared with the control group

**Table 3. Statistical distribution of the the cannabis user group [n = 100] according to their substance use motives.**

| Motives for substance intake | Group II [n = 100] | |
|---|---|---|
| | No. | % |
| **Frustration** | 21 | 21 |
| **Influence of friends** | 30 | 30 |
| **Lifestyle stress** | 33 | 33 |
| **Relationship difficulties** | 16 | 16 |

Group II–Cannabis group

**Table 4. Tetrahydrocannabinol level and Cannabis Withdrawal Assessment Scale [CWAS] for the the cannabis user group [n = 100] at admission.**

| Items | Group II [n = 100] |
|---|---|
| **Tetrahydrocannabinol level [ng/ml]** | 96.94 ± 2.60 |
| **CWAS** | 13.43 ± 5.48 |

CWAS- Cannabis Withdrawal Assessment Scale.

[p = < 0.05]. After treatment, the mean heart rate values demonstrated a significant decrease in the cannabis user group compared with the corresponding admission values [p = < 0.05].

In this study, Fig 2 and Table 5 showed a significant increase in the mean SBP values in the cannabis user group at admission and no significant change after treatment in the cannabis user group compared with the control group [p = < 0.05]. The mean SBP values after treatment in the cannabis user group demonstrated a significant decrease compared with the corresponding admission values [p = < 0.05].

The mean DBP values in the cannabis user group at admission and after treatment showed a significant increase compared with the control group [p = < 0.05]. The mean DBP values after treatment demonstrated a significant decrease in the cannabis user group compared with the corresponding admission values [p = < 0.05] [Fig 2 and Table 5].

As shown in Fig 2 and Table 5, there was a significant increase in the mean RR values in the cannabis user group at admission and no significant change in the cannabis user group after treatment compared with the control group [p = < 0.05]. The mean RR values after treatment demonstrated a significant decrease in the cannabis user group compared with the corresponding admission values [p = < 0.05].

The present study showed a significant increase in the mean albumin values in the cannabis user group at admission and no significant change in the cannabis user group after treatment compared with the control group [p = < 0.05]. The mean albumin values after treatment demonstrated a significant decrease in the cannabis user group compared with the corresponding at admission values [p = < 0.05] [Fig 3 and Table 5].

As shown in Fig 3 and Table 5, there was a significant increase in the mean total bilirubin values in the cannabis user group at admission and no significant change in the cannabis user group after treatment compared with the control group [p = < 0.05]. The mean total bilirubin values after treatment demonstrated a significant decrease in the cannabis user group compared with the corresponding at admission values [p = < 0.05].

The present study demonstrated a significant increase in the mean direct bilirubin values in the cannabis user group at admission and no significant change in the cannabis user group after treatment compared with the control group [p = < 0.05]. The mean direct bilirubin values after treatment demonstrated a significant decrease in the cannabis user group compared with the corresponding at admission values [p = < 0.05] [Fig 3 and Table 5].

This study demonstrated a significant increase in the mean aspartate aminotransferase values in the cannabis user group at admission and no significant change in the cannabis user group after treatment compared with the control group [p = < 0.05]. The mean aspartate aminotransferase values after treatment demonstrated a significant decrease in the cannabis user group compared with the corresponding at admission values [p = < 0.05] [Fig 3 and Table 5].

The present study demonstrated a significant increase in the mean alanine aminotransferase values in the cannabis user group at admission and no significant change in the cannabis user group after treatment compared with the control group [p = < 0.05]. The mean alanine

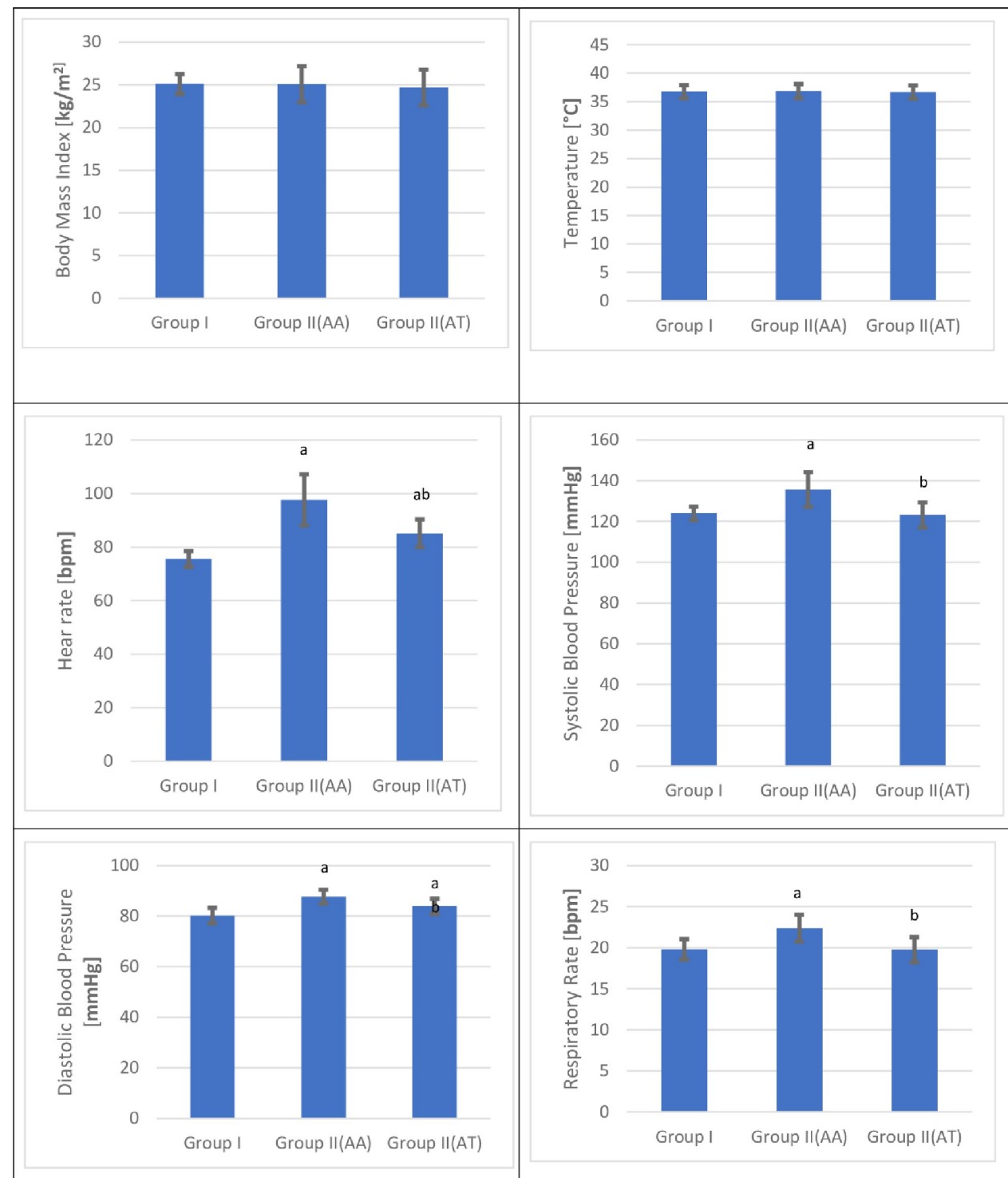

**Fig 2. Mean body mass index, temperature, heart rate, systolic and diastolic blood pressure, and respiratory rate values for the control and the cannabis user group s [n = 120] at admission [AA] and after treatment [AT].**

**Table 5. Body mass index, vital signs, Liver function tests, blood glucose level, kidney function tests and oxidative stress biomarkers for the control and the cannabis user group s [n = 120] before and after treatment.**

| Items | Group I [n = 20] | Group II [n = 100] | | P-value |
|---|---|---|---|---|
| | | **Before treatment** | **After treatment** | |
| Body Mass Index [kg/m²] | 25.09 ± 1.16 | 25.07 ± 2.12 | 24.69 ± 2.10 | 0.385 |
| Temperature [˚C] | 36.76 ± 1.16 | 36.86 ± 1.24 | 36.68 ± 1.18[b] | 0.574 |
| Heart Rate [bpm] | 75.50 ± 2.93 | 97.60 ± 9.51[a] | 85.06 ± 5.25[ab] | ≤0.001[#] |
| Systolic Blood Pressure [mmHg] | 123.90 ± 3.29 | 135.59 ± 8.56[a] | 123.21 ± 6.20[b] | ≤0.001[#] |
| Diastolic Blood Pressure [mmHg] | 80.20 ± 3.04 | 87.64 ± 2.78[a] | 83.87 ± 2.96[ab] | ≤0.001[#] |
| Respiratory Rate [bpm] | 19.80 ± 1.24 | 22.36 ± 1.65[a] | 19.75 ± 1.53[b] | ≤0.001[#] |
| Albumin [g/dL] | 4.46 ± 0.21 | 4.78 ± 0.55[a] | 4.40 ± 0.53[b] | ≤0.001[#] |
| Total bilirubin [mg/dL] | 0.92 ± 0.07 | 1.46 ± 0.36[a] | 0.96 ± 0.21[b] | ≤0.001[#] |
| Direct bilirubin [mg/dL] | 0.15 ± 0.07 | 0.23 ± 0.07[a] | 0.14 ± 0.08[b] | ≤0.001[#] |
| Aspartate aminotransferase [U/L] | 20.80 ± 5.80 | 32.57 ± 8.31[a] | 22.98 ± 3.82[b] | ≤0.001[#] |
| Alanine aminotransferase [U/L] | 20.25 ± 5.95 | 32.59 ± 8.24[a] | 23.24 ± 8.26[b] | ≤0.001[#] |
| Alkaline phosphatase [U/L] | 74.30 ± 18.24 | 90.30 ± 20.31[a] | 81.85 ± 25.76[b] | 0.003 |
| Fasting blood glucose level [mg/dL] | 88.00 ± 4.21 | 97.17 ± 13.71[a] | 84.78 ± 9.24[b] | ≤0.001[#] |
| Urea [mg/dL] | 30.33 ± 6.01 | 30.14 ± 8.59 | 30.60 ± 7.89 | 0.922 |
| Creatinine [mg/dL] | 0.90 ± 0.08 | 1.03 ± 0.20[a] | 1.02 ± 0.16[a] | 0.009 |
| Thiobarbituric acid-reactive substances [nmol/ml] | 1.23 ± 0.12 | 2.45 ± 0.17[a] | 2.43 ± 0.18[a] | ≤0.001[#] |
| Reduced glutathione [ng/ml] | 47.38 ± 2.21 | 28.63 ± 4.45[a] | 28.62 ± 3.77[a] | ≤0.001[#] |
| Superoxide dismutase [ng/ml] | 0.82 ± 0.08 | 1.96 ± 0.09[a] | 1.94 ± 0.12[a] | ≤0.001[#] |
| Catalase [ng/L] | 72.15 ± 2.22 | 57.25 ± 2.11[a] | 56.53 ± 4.37[a] | ≤0.001[#] |

Group I–Control group; Group II–Cannabis group, Values expressed as Mean ± SD.

[a] Significant compared with control group.

[b] Significant compared with before treatment Significant at p<0.05 level.

aminotransferase values after treatment demonstrated a significant decrease in group II compared with the corresponding at admission values [p = < 0.05] [Fig 3 and Table 5].

The current study demonstrated a significant increase in the mean alkaline phosphatase values in the cannabis user group at admission and no significant change in the cannabis user group after treatment compared with the control group [p = < 0.05]. The mean alkaline phosphatase values after treatment demonstrated a significant decrease in the cannabis user group compared with the corresponding at admission values [p = < 0.05] [Fig 3 and Table 5].

The present study demonstrated a significant increase in the mean fasting blood glucose level values in the cannabis user group at admission and no significant change in the cannabis user group after treatment compared with the control group [p = < 0.05]. The mean fasting blood glucose level values after treatment demonstrated a significant decrease in the cannabis user group compared with the corresponding at admission values [p = < 0.05] [Fig 3 and Table 5].

## Serum urea level

This study demonstrated no significant change in the mean urea values at admission and after treatment in the cannabis user group compared with the control group [p < 0.05]. The mean urea values after treatment demonstrated no significant change in the cannabis user group compared with the corresponding admission values [p < 0.05] [Fig 4 and Table 5].

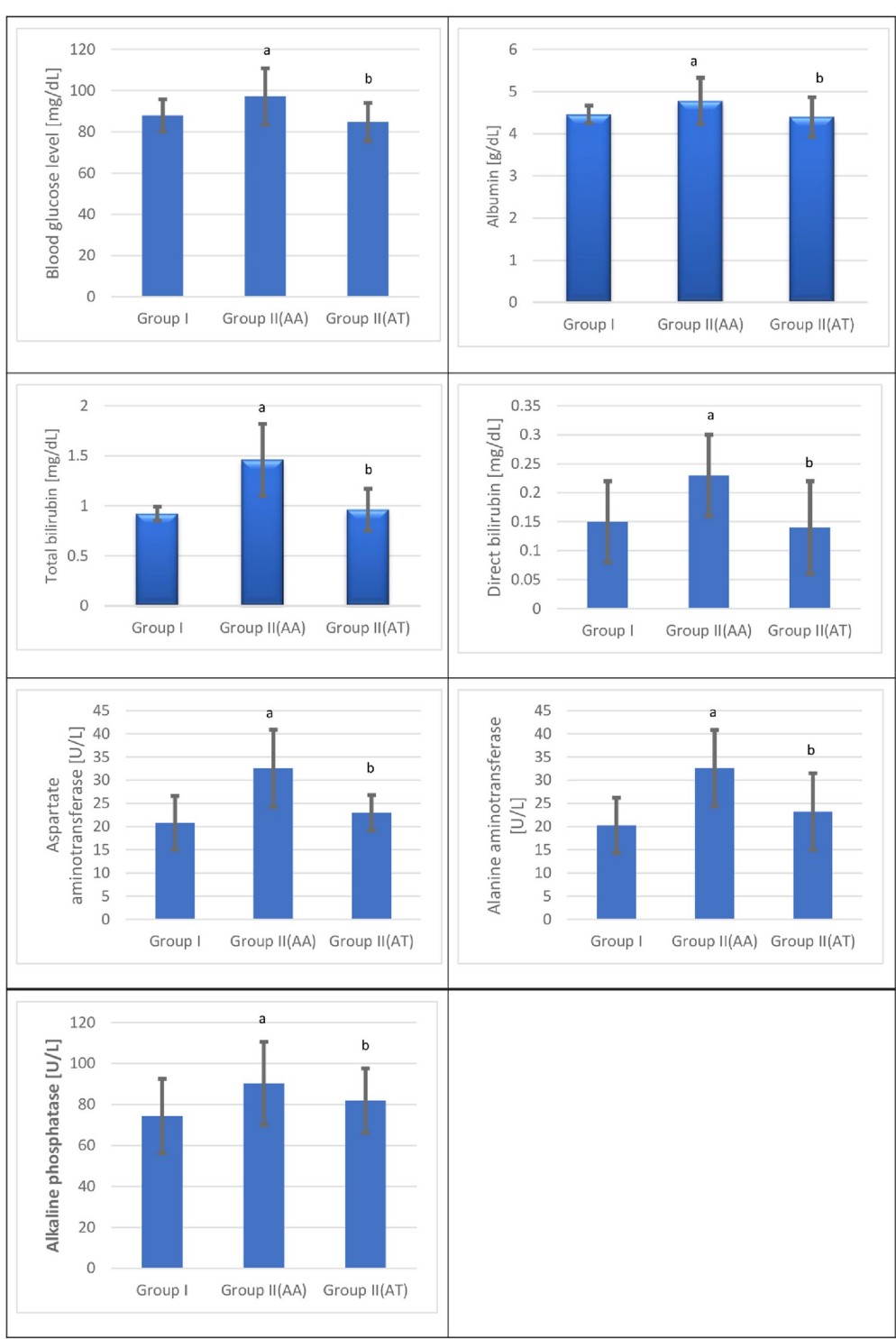

**Fig 3. Blood glucose level, albumin total bilirubin, direct bilirubin, Aspartate aminotransferase, Alanine aminotransferase, and Alkaline phosphatase for the control and the cannabis user group s [n = 120] at admission [AA] and after treatment [AT].**

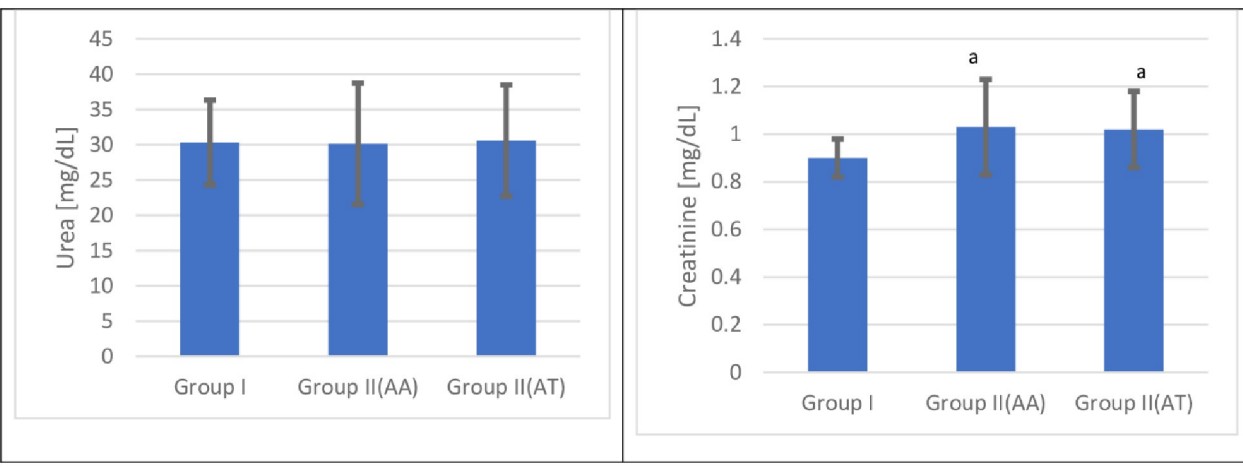

**Fig 4. Serum urea and creatinine values for the control and the cannabis user group s [n = 120] at admission [AA] and after treatment [AT].**

### Serum creatinine level

The present study showed a significant increase in the mean creatinine values at admission and after treatment in the cannabis user group compared with the control group [p < 0.05]. The mean creatinine values after treatment demonstrated no significant change in the cannabis user group compared with the corresponding admission values [p < 0.05] [Fig 4 and Table 5].

The present study demonstrated a significant increase in the mean thiobarbituric acid-reactive substances values in the cannabis user group at admission and after treatment compared with the control group [p < 0.05]. The mean thiobarbituric acid-reactive substances values after treatment demonstrated no significant change in the cannabis user group compared with the corresponding admission values [p < 0.05] [Fig 5 and Table 5].

The present study demonstrated a significant decrease in the mean reduced glutathione values in the cannabis user group at admission and after treatment compared with the control group [p < 0.05]. The mean reduced glutathione values after treatment demonstrated no significant change in the cannabis user group compared with the corresponding admission values [p < 0.05] [Fig 5 and Table 5].

The present study demonstrated a significant increase in the mean superoxide dismutase values in the cannabis user group at admission and after treatment compared with the control group [p < 0.05]. The mean superoxide dismutase values after treatment demonstrated no significant change in the cannabis user group compared with the corresponding admission values [p < 0.05] [Fig 5 and Table 5].

The present study demonstrated a significant decrease in the mean catalase values in group II at admission and after treatment compared with the control group [p < 0.05]. The mean catalase values after treatment demonstrated no significant change in the cannabis user group compared with the corresponding admission values [p < 0.05] [Fig 5 and Table 5].

### Correlations of the studied parameters for young patients [aged 18 to 35 years old]

**Body Mass Index [BMI].** Table [6] revealed that there were no significant correlations in the cannabis user group at admission and after treatment between BMI and CWAS, tetrahydrocannabinol level, age and duration of cannabis intake.

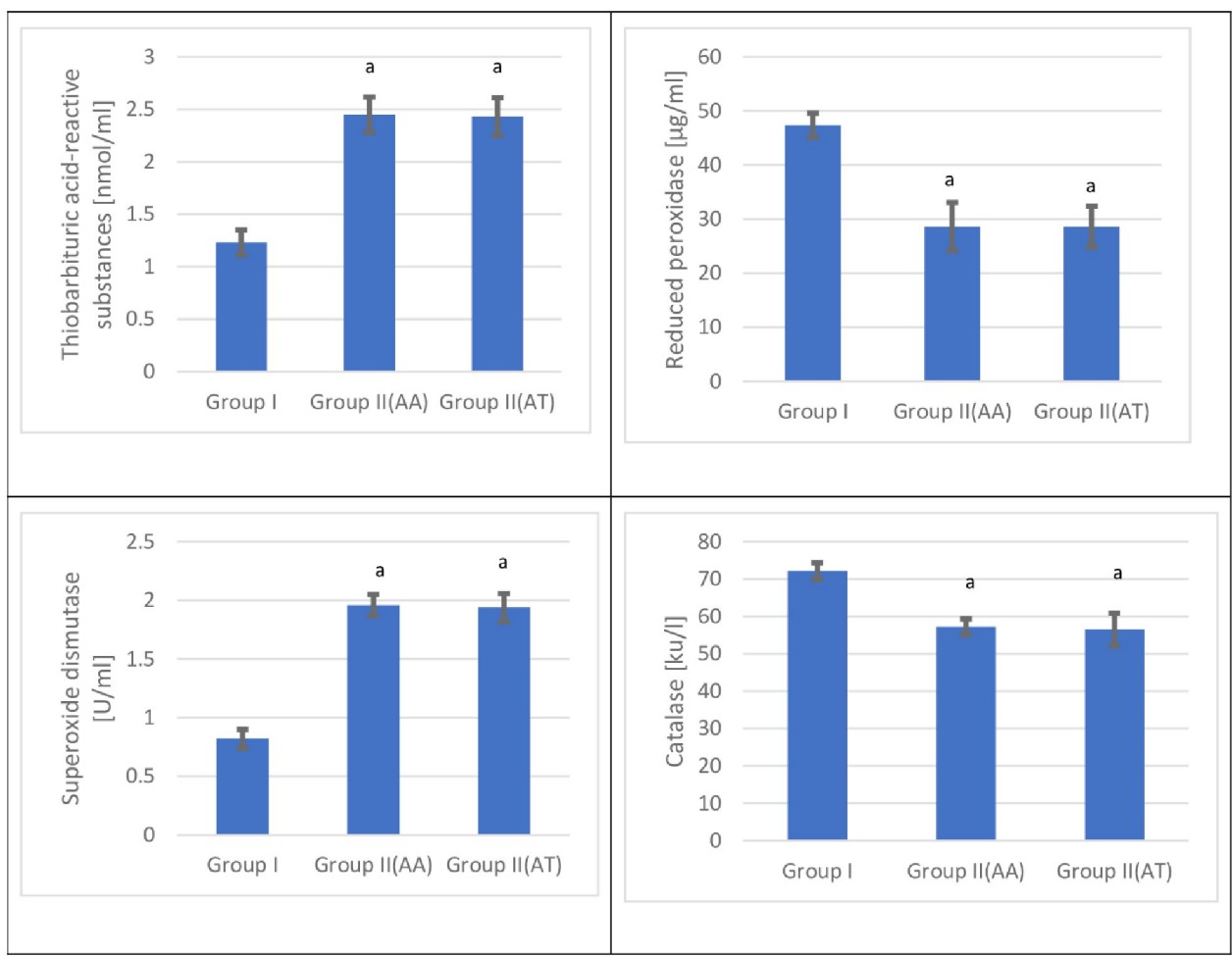

**Fig 5. Serum thiobarbituric acid-reactive substances, reduced glutathione, superoxide dismutase, and catalase values for the control and the cannabis user group s [n = 120] at admission [AA] and after treatment [AT].**

**Table 6. Pearson correlation coefficients between Cannabis Withdrawal Assessment Scale [CWAS], cannabis level, age and duration of cannabis intake, body mass index, and vital signs in the cannabis group at admission [AA] and after the treatment [AT]] for patients aged 18 to 35 years old, [n = 59].**

| Items | CWAS | | 9-THC level | | Age | | Duration of cannabis intake | |
|---|---|---|---|---|---|---|---|---|
| | AA | AT | AA | AT | AA | AT | AA | AT |
| BMI [kg/m²] | 0.099 | 0.125 | 0.087 | 0.083 | 0.096 | 0.045 | 0.067 | 0.098 |
| Temp. [˚C] | 0.043 | 0.098 | 0.078 | 0.055 | 0.046 | 0.076 | 0.052 | 0.049 |
| HR [bpm] | 0.082 | 0.262 | 0.793** | 0.110 | 0.098 | 0.065 | 0.578** | 0.048 |
| SBP [mmHg] | 0.134 | 0.017 | 0.728** | 0.146 | 0.698** | 0.079 | 0.799** | 0.112 |
| DBP [mmHg] | 0.067 | 0.092 | 0.056 | 0.076 | 0.077 | 0.098 | 0.069 | 0.066 |
| RR [bpm] | 0.067 | 0.093 | 0.083 | 0.089 | 0.056 | 0.058 | 0.089 | 0.049 |

CWAS—Cannabis Withdrawal Assessment Scale; 9-THC- tetrahydrocannabinol; Temp- Temperature; BMI- Body Mass Index; HR- Heart Rate; SBP–Systolic Blood Pressure; DBP–Diastolic Blood Pressure; RR- Respiratory Rate.

* Correlation was statistically significant at 0.05 level [2-tailed].

** Correlation was statistically significant at 0.01 level [2-tailed].

**Vital signs.** Table [6] revealed that body temperature had no significant correlation in the cannabis user group before and after treatment with CWAS, tetrahydrocannabinol level, age, and the duration of cannabis intake.

Table [6] revealed that HR had no significant correlation in the cannabis user group at admission and after treatment with CWAS and age. In comparison, HR showed significant correlation in group II at admission with tetrahydrocannabinol level and the duration of cannabis intake and no significant correlation in the cannabis user group after treatment with tetrahydrocannabinol level and the duration of cannabis intake.

Table [6] revealed that SBP had significant correlations in the cannabis user group at admission with tetrahydrocannabinol level, age, and the duration of cannabis intake and no significant correlations after treatment. SBP showed no significant correlation in the cannabis user group at admission and after treatment with CWAS.

Table [6] revealed that DBP had no significant correlation in the cannabis user group at admission and after treatment with CWAS, tetrahydrocannabinol level, age, and duration of cannabis intake.

Table [6] revealed that there were no significant correlations in the cannabis user group at admission and after treatment between RR and CWAS, tetrahydrocannabinol level, age, and duration of cannabis intake.

**Liver function tests.** Table [7] revealed that there were no significant correlations in the cannabis user group at admission and after treatment between albumin and CWAS, tetrahydrocannabinol level, age, and the duration of cannabis intake.

Table [7] revealed that there were no significant correlations in the cannabis user group at admission and after treatment between total and direct bilirubin and CWAS, tetrahydrocannabinol level, age, and the duration of cannabis intake.

Table [7] revealed that there were significant correlations in the cannabis user group at admission between aspartate aminotransferase and alanine aminotransferase and tetrahydrocannabinol level and the duration of cannabis intake. On the other hand, they showed no significant correlations after treatment. While, they showed no significant correlations between CWAS and age at admission and after treatment.

**Table 7. Pearson correlation coefficients between Cannabis Withdrawal Assessment Scale [CWAS], tetrahydrocannabinol level, age, duration of cannabis intake, liver function tests values, and blood glucose level in the cannabis group at admission [AA] and after the treatment [AT]] for patients aged 18 to 35 years old, [n = 59].**

| Items | CWAS | | 9-THC level | | Age | | Duration of cannabis intake | |
|---|---|---|---|---|---|---|---|---|
| | AA | AT | AA | AT | AA | AT | AA | AT |
| Albumin [g/dL] | 0.099 | 0.076 | 0.088 | 0.067 | 0.058 | 0.049 | 0.099 | 0.086 |
| Total bilirubin [mg/dL] | 0.082 | 0.079 | 0.096 | 0.068 | 0.034 | 0.047 | 0.067 | 0.084 |
| Direct bilirubin [mg/dL] | 0.045 | 0.055 | 0.078 | 0.089 | 0.094 | 0.088 | 0.064 | 0.062 |
| AST [U/L] | 0.056 | 0.052 | 0.720** | 0.112 | 0.114 | 0.133 | 0.510* | 0.101 |
| ALT [U/L] | 0.122 | 0.048 | 0.778** | 0.126 | 0.046 | 0.054 | 0.599** | 0.076 |
| ALP [U/L] | 0.054 | 0.066 | 0.110 | 0.119 | 0.077 | 0.098 | 0.069 | 0.100 |
| Blood glucose level [mg/dL] | 0.086 | 0.098 | 0.105 | 0.100 | 0.079 | 0.048 | 0.058 | 0.089 |

CWAS- Cannabis Withdrawal Assessment Scale; 9-THC- tetrahydrocannabinol; AST- Aspartate aminotransferase; ALT- Alanine aminotransferase; ALP- Alkaline phosphatase.

* Correlation was statistically significant at 0.05 level [2-tailed].

** Correlation was statistically significant at 0.01 level [2-tailed].

Table [7] revealed that there were no significant correlations in the cannabis user group at admission and after treatment between alkaline phosphatase and CWAS, tetrahydrocannabinol level, age, and the duration of cannabis intake.

Table [7] revealed that there were significant correlations in the cannabis user group at admission and after treatment between blood glucose level and CWAS, tetrahydrocannabinol level, age, and duration of cannabis intake.

Table [8] revealed that there were no significant correlations in the cannabis user group at admission and after treatment between urea and CWAS, tetrahydrocannabinol level, age, and duration of cannabis intake.

Table [8] revealed that there were no significant correlations in the cannabis user group at admission and after treatment between creatinine and CWAS, tetrahydrocannabinol level, age, and duration of cannabis intake.

Table [9] revealed that there were significant correlations in the cannabis user group at admission between oxidative stress markers [Thiobarbituric acid-reactive substances, reduced glutathione, superoxide dismutase and catalase] and tetrahydrocannabinol leveland the duration of cannabis intake. On the other hand, they showed no significant correlations after treatment. While, they showed no significant correlations with CWAS and age at admission and after treatment.

## Correlations of the studied parameters for patients aged more than 35 years old

**Body Mass Index [BMI].**   Table [10] revealed that there were no significant correlations in the cannabis user group at admission and after treatment between BMI and age, duration of cannabis intake and the tetrahydrocannabinol level. On the other hand, the present study revealed that there were significant correlations in the cannabis user group at admission and after treatment between BMI and duration of cannabis intake.

**Vital signs.**   Table [10] revealed that body temperature had no significant correlation in the cannabis user group before and after treatment with CWAS, tetrahydrocannabinol level, age, and duration of cannabis intake.

Table [10] revealed that HR had no significant correlation in the cannabis user group at admission and after treatment with CWAS. While, HR showed significant correlations in group II at admission and after treatment with tetrahydrocannabinol level, age and the duration of cannabis intake.

Table [10] revealed that SBP had no significant correlations in the cannabis user group at admission and after treatment with CWAS. Whereas, SBP showed significant correlations in group II at admission and after treatment with the tetrahydrocannabinol level, age and the duration of cannabis intake.

**Table 8. Pearson correlation coefficients between Cannabis Withdrawal Assessment Scale [CWAS], tetrahydrocannabinol level, age, duration of cannabis intake, renal function tests values in the cannabis group at admission [AA] and after the treatment [AT]] for patients aged 18 to 35 years old, [n = 59].**

| Items | CWAS | | 9-THC level | | Age | | Duration of cannabis intake | |
|---|---|---|---|---|---|---|---|---|
| | AA | AT | AA | AT | AA | AT | AA | AT |
| Urea [mg/dL] | 0.121 | 0.111 | 0.067 | 0.099 | 0.114 | 0.115 | 0.088 | 0.076 |
| Creatinine [mg/dL] | 0.043 | 0.034 | 0.055 | 0.064 | 0.039 | 0.046 | 0.052 | 0.064 |

CWAS -Cannabis Withdrawal Assessment Scale; 9-THC- tetrahydrocannabinol.

* Correlation was statistically significant at 0.05 level [2-tailed].

** Correlation was statistically significant at 0.01 level [2-tailed].

**Table 9. Pearson correlation coefficients between CWAS, tetrahydrocannabinol level, age, duration of cannabis, enzymatic antioxidants, and lipid peroxidation in the cannabis group at admission [AA] and after the treatment [AT]] for patients aged 18 to 35 years old, [n = 59].**

| Items | CWAS | | 9-THC level | | Age | | Duration of cannabis intake | |
|---|---|---|---|---|---|---|---|---|
| | AA | AT | AA | AT | AA | AT | AA | AT |
| TBARS | 0.111 | 0.091 | 0.708** | 0.101 | 0.069 | 0.099 | 0.858** | 0.098 |
| GSH [mg/dl] | -0.004 | -0.071 | -0.657** | -0.117 | -0.112 | -0.057 | -0.718** | -0.109 |
| SOD [ng/ml] | 0.048 | 0.067 | 0.625** | 0.095 | 0.096 | 0.078 | 0.365** | 0.065 |
| CAT [ku/l] | -0.098 | -0.112 | - 0.546** | -0.112 | -0.078 | -0.087 | -0.495** | -0.088 |

CWAS -Cannabis Withdrawal Assessment Scale; 9-THC- tetrahydrocannabinol; TBARS–Thiobarbituric acid-reactive substances; GSH–Reduced glutathione; SOD–Superoxide dismutase; CAT–Catalase.

\* Correlation was statistically significant at 0.05 level [2-tailed].

\*\* Correlation was statistically significant at 0.01 level [2-tailed].

Table [10] revealed that DBP had no significant correlation in the cannabis user group at admission and after treatment with CWAS, tetrahydrocannabinol level, age, and duration of cannabis intake.

Table [10] revealed that there were significant correlations in the cannabis user group at admission and after treatment between RR and the duration of cannabis intake. On the other hand, the present study revealed that there were no significant correlations in the cannabis user group at admission and after treatment between RR and the tetrahydrocannabinol levels, CWAS and age.

**Liver function tests.** Table [10] revealed that there were no significant correlations in the cannabis user group at admission and after treatment between albumin and CWAS, tetrahydrocannabinol level, age, and the duration of cannabis intake.

Table [11] revealed that there were no significant correlations in the cannabis user group at admission and after treatment between total and direct bilirubin and CWAS, the tetrahydrocannabinol levels, age, and duration of cannabis intake.

Table [11] revealed that there were significant correlations in the cannabis user group at admission and after treatment between liver enzymes [aspartate aminotransferase and alanine aminotransferase] and the tetrahydrocannabinol level, age, and the duration of cannabis intake. On the other hand, they showed no significant correlations in the cannabis user group

**Table 10. Pearson correlation coefficients between Cannabis Withdrawal Assessment Scale [CWAS], cannabis level, age and duration of cannabis intake, body mass index, and vital signs in the cannabis group at admission [AA] and after the treatment [AT] for patients aged more than 35 years old, [n = 41].**

| Items | CWAS | | 9-THC level | | Age | | Duration of cannabis intake | |
|---|---|---|---|---|---|---|---|---|
| | AA | AT | AA | AT | AA | AT | AA | AT |
| BMI [kg/m$^2$] | 0.089 | 0.125 | 0.055 | 0.045 | 0.067 | 0.062 | 0.689** | 0.576** |
| Temp. [˚C] | 0.082 | 0.111 | 0.119 | 0.067 | 0.083 | 0.084 | 0.092 | 0.043 |
| HR [bpm] | 0.082 | 0.262 | 0.793** | 0.266* | 0.749** | 0.386** | 0.454** | 0.267* |
| SBP [mmHg] | 0.124 | 0.099 | 0.896** | 0.256* | 0.844** | 0.239* | 0.779** | 0.355** |
| DBP [mmHg] | 0.087 | 0.098 | 0.115 | 0.053 | 0.114 | 0.067 | 0.118 | 0.052 |
| RR [bpm] | 0.056 | 0.092 | 0.085 | 0.037 | 0.056 | 0.067 | 0.754** | 0.398** |

CWAS—Cannabis Withdrawal Assessment Scale; 9-THC- tetrahydrocannabinol; Temp- Temperature; BMI- Body Mass Index; HR- Heart Rate; SBP–Systolic Blood Pressure; DBP–Diastolic Blood Pressure; RR- Respiratory Rate.

\* Correlation was statistically significant at 0.05 level [2-tailed].

\*\* Correlation was statistically significant at 0.01 level [2-tailed].

**Table 11. Pearson correlation coefficients between Cannabis Withdrawal Assessment Scale [CWAS], tetrahydrocannabinol level, age, duration of cannabis intake, liver function tests values, and blood glucose level in the cannabis group at admission [AA] and after the treatment [AT] for patients aged more than 35 years old, [n = 41].**

| Items | CWAS | | 9-THC level | | Age | | Duration of cannabis intake | |
|---|---|---|---|---|---|---|---|---|
| | AA | AT | AA | AT | AA | AT | AA | AT |
| Albumin [g/dL] | 0.129 | 0.166 | 0.091 | 0.026 | 0.067 | 0.092 | 0.127 | 0.158 |
| Total bilirubin [mg/dL] | 0.082 | 0.079 | 0.106 | 0.099 | 0.021 | 0.029 | 0.089 | 0.094 |
| Direct bilirubin [mg/dL] | 0.023 | 0.007 | 0.115 | 0.071 | 0.102 | 0.082 | 0.085 | 0.042 |
| AST [U/L] | 0.094 | 0.105 | 0.620** | 0.205* | 0.512** | 0.200* | 0.710** | 0.398** |
| ALT [U/L] | 0.191 | 0.142 | 0.479** | 0.321** | 0.381** | 0.233* | 0.679** | 0.359** |
| ALP [U/L] | 0.093 | 0.094 | 0.110 | 0.119 | 0.078 | 0.102 | 0.108 | 0.131 |
| Blood glucose level [mg/dL] | 0.106 | 0.094 | 0.605** | 0.200* | 0.673** | 0.356** | 0.648** | 0.383** |

CWAS- Cannabis Withdrawal Assessment Scale; 9-THC- tetrahydrocannabinol; AST- Aspartate aminotransferase; ALT- Alanine aminotransferase; ALP- Alkaline phosphatase.

* Correlation was statistically significant at 0.05 level [2-tailed].

** Correlation was statistically significant at 0.01 level [2-tailed].

at admission and after treatment with CWAS. The current study revealed that there were no significant correlations in the cannabis user group at admission and after treatment between alkaline phosphatase and CWAS, tetrahydrocannabinol levels, age, and duration of cannabis intake.

Table [11] revealed that there were significant correlations in the cannabis user group at admission and after treatment between blood glucose level and tetrahydrocannabinol level, age, and duration of cannabis intake. On the other hand, it showed no significant correlations in the cannabis user group at admission and after treatment with CWAS.

Table [12] revealed that there were no significant correlations in the cannabis user group at admission and after treatment between urea and CWAS, tetrahydrocannabinol level, age, and duration of cannabis intake.

Table [12] revealed that there were no significant correlations in the cannabis user group at admission and after treatment between creatinine and CWAS, tetrahydrocannabinol level, age, and duration of cannabis intake.

## Discussion

This work evaluates biological (including oxidative stress) markers and vital signs of intervention for treating cannabis addiction. It demonstrated that most of the biological and vital signs

**Table 12. Pearson correlation coefficients between Cannabis Withdrawal Assessment Scale [CWAS], tetrahydrocannabinol level, Age, duration of Cannabis intake, renal function tests values in the cannabis group at admission [AA] and after the treatment [AT] for patients aged more than 35 years old, [n = 41].**

| Items | CWAS | | 9-THC level | | Age | | Duration of cannabis intake | |
|---|---|---|---|---|---|---|---|---|
| | AA | AT | AA | AT | AA | AT | AA | AT |
| Urea [mg/dL] | 0.131 | 0.161 | 0.084 | 0.100 | 0.145 | 0.142 | 0.096 | 0.097 |
| Creatinine [mg/dL] | 0.029 | 0.028 | -0.048 | -0.034 | 0.035 | 0.025 | 0.047 | 0.058 |

CWAS -Cannabis Withdrawal Assessment Scale; 9-THC- tetrahydrocannabinol.

* Correlation was statistically significant at 0.05 level [2-tailed].

** Correlation was statistically significant at 0.01 level [2-tailed].

alterations were resolved compared to the control group. In contrast, oxidative stress markers were still high.

The primary predisposing factors for initiating substance consumption and thus abuse were peer pressure, family influence, and anxiety and stress reduction. Information and studies on the epidemiology of drug abuse in Saudi Arabia are relatively limited [28]. It is important to note that the lack of female participants in this study may be attributable to the strict socio-cultural environment surrounding the cases' confidentiality. Recently, this restriction was reported in the Qassim region [7].

Compared to adults, adolescent Cannabis use was found to be more prevalent. Numerous studies have determined that the prevalence of Cannabis use varies by age and region, ranging from 1 to 60 percent. Eleven percent of adults and up to twenty percent of adolescents reported using Cannabis [4]. Adults and adolescents begin substance abuse with tobacco, amphetamine, and inhalants, according to another study conducted in Jeddah to determine the progression of drug abusers, Cannabis is presented. It was consumed at a rate that was twice that of adults by adolescents. Both adults and adolescents increased their consumption by 37% and 57%, respectively [6]. Peer pressure was the primary cause of substance abuse among young men, as it was the most prevalent factor [29]. These reports confirm females' special circumstances surrounding evaluating substance use disorders [SUD]. Saquib et al. [2020] reported that the male gender was a significant risk factor for substance use disorder in KSA. This could be because males have more freedom than females to leave the house and stay out late at night, visit rest areas, and vacation with their peers. [28]. In addition to being a significant problem, substance use disorder is associated with educational and occupational consequences and socioeconomic burden [4].

This study indicates that the age group most affected by Cannabis-related disorders is between 26 and 35. Multiple reports indicated a rise in SUD among the Saudi populace. Almost 8% of Saudis have admitted to using illegal substances at some point [28]. Following these findings, a global report revealed an upward trend in substance abuse among young users, with amphetamine and Cannabis being the most frequently abused substances [7]. In addition, most drug addicts were found in urban areas in the present study. This is consistent with the results found by Coughlin et al., 2019 [30]. In contrast, Reilly et al. [1998] demonstrated that Cannabis use was an integral part of daily life among long-term Cannabis users in rural Australia [31].

This study revealed that most participants of the Cannabis user group were unmarried or divorced. Saquib et al. reported in 2020, respectively, that single status was associated with dependence on Cannabis and other stimulants. These studies on Cannabis consumers supported our findings. Most participants lived with their friends, with varying percentages living with their families or alone due to circumstances such as living far away from their homes and working or attending school [28].

Most the Cannabis user group participants had either a bachelor's degree or high school education, while a low net monthly income characterized their employment status. In 2017, a case-control study analyzed the risk factors among female drug users and found that SUD patients were more likely than controls to be unemployed, have unstable marriages, have less education, have low family income, and have unstable family conditions [32]. Ibrahim et al. conducted a study at the Qassim rehabilitation center that was consistent with the present study's findings [7].

In addition, the highest proportion of patients were heavy smokers, and the majority of Cannabis users had consumed the drug for between two and five years. An identical trend was reported in Qassim [7]. Furthermore, we found that the influence of friends and the work

environment play a significant role in using illicit drugs. Most drug users were unemployed, which may be linked to their use of substances to escape the reality of their situation [33].

Energy uptake, storage, and conservation are primarily mediated by CB1 receptors. Through several techniques, it works to maximize energy uptake and conservation. Food is made more palatable by modulating taste and smell pathways by stimulating CB1 receptors [34]. There is much proof that THC or cannabis use causes the CB1R to be downregulated. Regular cannabis usage is linked to CB1R downregulation and desensitization, and CB1R levels stay low for 3–4 weeks after use has stopped [35]. During this time, recently abstinent users would have decreased hunger and elevated metabolic rates. They will not, however, continue to have short-term stimulation of appetite, increased energy consumption, and decreased metabolic rates. Therefore, until CB1R levels reach those from before cannabis usage, the weight will not rise because energy intake and storage are still suppressed, and metabolism is increased [36]. Ngueta et al., 2015 reported that a higher metabolic rate in marijuana users explains the lack of association between cannabis use and BMI, despite a high-energy diet [37]. These data explained the findings of the current study. Compared to the control group, the current results do not indicate a significant reduction in BMI at admission and after treatment.

Immediately following marijuana use, the heart rate and blood pressure may increase [38]. Increased sympathetic nervous system activity is responsible for the increased heart rate, supine systolic and diastolic blood pressures, and forearm blood flow caused by marijuana cigarettes [39]. Tolerance to many of these effects is first evident after a few repeated doses. Tolerance to increased heart rate and blood pressure changes can be profound after only a day or two of frequent exposure. When marijuana or THC is stopped, tolerance is lost rapidly. After a few days of repeated exposure to frequent doses of oral THC or a marijuana plant extract, blood pressure decreases, and the resting heart rate slows [40]. The current study results revealed a significant increase in the mean heart rate values of the cannabis user group at admission and after treatment compared to the control group. In addition, the mean heart rate values after treatment were significantly lower in the cannabis user group after treatment compared to the corresponding admission values.

In the present study, the mean SBP and DBP values were significantly lower in the cannabis user group after treatment than at admission values. In addition, Kichloo et al., 2021 discovered a significant increase in systolic and diastolic blood pressure at 10 mg doses of marijuana. Changes in blood pressure appear to be secondary to an elevated heart rate rather than the marijuana dosage [41]. In contrast, Cannabis use was associated with a lower BMI and a decreased risk of cardiometabolic risk factors. These results must be interpreted in light of prior research indicating that Cannabis use increases the risk of cardiovascular events, accidents and injuries, persistent psychotic symptoms, schizophrenia, and psychosocial issues [37].

The current study demonstrated that the mean RR values significantly increased in the Cannabis user group at admission, and there was no significant change after treatment compared with the control group. Long-term marijuana smoking and airflow obstruction have a pathophysiologic relationship that is still unknown. Regular Cannabis use, on the other hand, has been linked to an increased risk of asthma, COPD, and pneumonia [38]. In addition, Chronic cannabis users are more likely to experience respiratory problems like chronic coughing, sputum production, dyspnea, hoarseness, chest tightness, and lung cancer [39].

There was no significant difference in mean temperature values at admission and after treatment for the cannabis user group when compared to those of the control group. Rectal temperature and most peripheral temperatures were not significantly altered by marijuana smoking [40].

The present study demonstrated that, compared to the control group, there was a significant increase in the mean fasting blood glucose level values of the cannabis user group at admission but no significant change in the cannabis user group after treatment. After treatment, the mean fasting blood glucose levels in the cannabis user group decreased significantly compared to their respective admission values. Marijuana use was linked to lower fasting insulin levels and a homeostasis model assessment of insulin resistance [41]. Cannabis use for 12 months is linked to nearly double the risk of diabetic ketoacidosis in one study of adults with T1D compared to no cannabis use [42]. The potency variation of the cannabis types could explain the fluctuation in blood glucose levels. The inadequate preparation ('bhang') acts as an anorexigenic agent, while the potent preparations ('ganja', 'charas') inhibit appetite. In humans smoking one or more marijuana cigarettes, THC levels in peripheral blood exceeded the range of 5 to 250 ng/ml, while intravenous administration of 10 mg/kg THC resulted in a blood level of 720 ng/ml after 20 minutes [9]. Other studies have shown that cannabinoids increase adrenocorticotropin (ACTH) secretion [43] and are also involved in the stress-response [44].

When comparing the mean creatinine values after treatment to the corresponding admission values, the cannabis user group showed no significant change. According to Bonnet et al. 2021, chronic recreational Cannabis use did not affect kidney function. Furthermore, there was no critical link between GFR and reported Cannabis burden [median daily use of 2.5 g for 36 months, moderate general symptom load]. The GFR was also not significantly correlated with serum cannabinoid levels, which was surprising [45].

Hepatotoxicity is a possible side effect of using illegal drugs, such as marijuana. A study by Borini et al. evaluating clinical and laboratory changes in chronic marijuana users showed that about fifty percent of patients have an enlarged spleen, liver, or both. Also, marijuana use has been linked to increased serum levels of AST, ALT, and AP, regardless of the amount or duration of use [46]. Our findings found that compared to the control group, there was a significant increase in the mean levels of albumin, total bilirubin, direct bilirubin, AST, ALT, and ALP in the cannabis user group at admission but no significant change in the cannabis user group after treatment. These results align with the study of Devinsky et al., 2018 who found that chronic cannabis users exhibited increases in liver enzymes that normalized after stopping its administration [47]. Also, Wani et al., 2017 discovered that cannabis abusers had higher levels of total bilirubin, bilirubin direct, bilirubin indirect, aspartate amino transferase, alanine amino transferase, alkaline phosphate, and gamma-glutamyl transferase, and lower levels of total protein, albumin, and globulin, compared to the control group [48]. Borini et al., 2004 and Ewing et al., 2019 reported that Cannabis induces hepatotoxicity due to cholestatic injury, confirmed by elevation of ALP. The resolution of liver alterations could be explained by the fact that Cannabis is less likely to induce liver fibrosis [46, 49].

Many cellular antioxidant enzymes and molecules protect cells from the harmful effects of oxidative stress. These include enzymatic antioxidants such as superoxide dismutase, catalase, and glutathione peroxidase, as well as non-enzymatic antioxidants such as vitamin C, vitamin E and glutathione [50]. Whenever the capacity of the cellular antioxidant system is overwhelmed by excessive free radical generation, oxidative stress is said to be present, with consequent damage to membrane lipids, i.e., lipid peroxidation, and the oxidation of intracellular enzyme proteins and nucleic acids [51]. The excessive formation of reactive oxygen species [ROS] and impairment of defensive antioxidant systems lead to a condition known as oxidative stress. Mitochondrial dysfunction is the primary source of free radicals responsible for oxidative stress [52]. Cannabis has been described to induce mitochondrial dysfunction [53]. THC induced a significant production of ROS. Since mitochondria are both causes and targets of ROS, mitochondrial production of H2O2 might be increased due to cannabis-related mitochondrial dysfunction. Accordingly, the free radical leak increased after THC exposure,

supporting the fraction of electrons that reduce O2 to ROS in the respiratory chain [54]. In 2017, Bayazit et al. discovered higher levels of total oxidant status, oxidative stress index, interleukin [IL] 1, IL-6, IL-8, and tumor necrosis factor [TNF] in cannabis-dependent individuals compared to healthy individuals [55]. In 2020, Guler et al. concluded that SC and its potential for DNA damage could cause an increase in oxidative stress and inflammatory processes [56]. In 2016, Nagy et al. discovered a significant rise in both TBARS and protein oxidation in heroin and cannabis male and female addicts compared to the control group. They concluded that cannabis addiction harmed the immune system by affecting oxidative stress and ROS release [57]. These data are consistent with the findings of this study. The present study demonstrated a significant increase in the mean thiobarbituric acid-reactive substances, superoxide dismutase values and a decrease in the mean reduced glutathione and catalase values in the cannabis user group at admission compared to healthy controls. These findings agree with a previous report showing enhanced oxidative stress markers after using Cannabis [58]. Also, Cerretani et al., 2020 reported that the reduced glutathione/oxidized glutathione ratio and catalase activity were significantly reduced and increased in malondialdehyde content after exposure to Cannabis [59]. On the other hand, in 2021, Dawidowicz et al. found that THC has antioxidant properties similar to vitamins E and C, scavenging free radicals, reducing metal ions, and protecting oxidation processes [60]. Our study notes that the dose of Cannabis used has not been reported, and the timing of measurement is different in our study. As a result, various cellular adaptations are very likely to occur. Bani-Ahmad reported that TBARS concentration showed no significant change among hashish addicts compared with the control group. The insignificant oxidative markers levels among hashish addicts might be a result of the contribution of multiple factors that represent various aspects of hashish cultivation, preparation, and hashish pharmacokinetics through the smoking route. Collectively, these factors may contribute to limited and unpredicted plasma levels of potent THC metabolites [61].

## Limitations of the study

The limitations of this research were a cross-sectional design, a relatively short period of follow-up with the participant, and a small sample size. All the participants in this work were male. It needs a Longitudinal study with more examination and confirmation of the addictive components by coordinating with the Drug Control Department at the Ministry of Interior and following the necessary requirements. Then working on a prolonged program to treat a group of addicts through specialized advanced treatment centers.

## Conclusion

The current findings show that cannabis addiction harms various body organs except the kidneys and seriously affects addicts' health. This study revealed that after treatment, there was no improvement in oxidative stress biomarker values, but there was an improvement in the other biological parameters.

## Supporting information

**S1 Data.**
(XLSX)

## Author Contributions

**Conceptualization:** Mohammed Ali A. L. Siwar, Ashraf Mahmoud Emara.

**Data curation:** Mohammed Ali A. L. Siwar, Ashraf Mahmoud Emara.

**Formal analysis:** Ashraf Mahmoud Emara.

**Funding acquisition:** Mohammed Ali A. L. Siwar, Ashraf Mahmoud Emara.

**Investigation:** Mohammed Ali A. L. Siwar, Ashraf Mahmoud Emara.

**Methodology:** Mohammed Ali A. L. Siwar, Sayed Hossam El-Din Mansour, Ashraf Mahmoud Emara.

**Project administration:** Mohammed Ali A. L. Siwar, Ashraf Mahmoud Emara.

**Resources:** Ashraf Mahmoud Emara.

**Software:** Ashraf Mahmoud Emara.

**Supervision:** Ashraf Mahmoud Emara.

**Validation:** Ahmad H. Alhowail, Yasser S. Almogbel, Ashraf Mahmoud Emara.

**Visualization:** Ashraf Mahmoud Emara.

**Writing – original draft:** Mohammed Ali A. L. Siwar, Ashraf Mahmoud Emara.

**Writing – review & editing:** Maha A. Aldubayan, Ahmad H. Alhowail, Yasser S. Almogbel, Ashraf Mahmoud Emara.

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
