## [Decision Letter · Decision Letter 0]

17 Mar 2023

PONE-D-23-02435

Health status outcome among cannabis addicts after treatment of addiction

PLOS ONE

Dear Dr. Emara,

Thank you for submitting your manuscript to PLOS ONE. After careful consideration, we feel that it has merit but does not fully meet PLOS ONE’s publication criteria as it currently stands. Therefore, we invite you to submit a revised version of the manuscript that addresses the points raised during the review process.

We look forward to receiving your revised manuscript.

Kind regards,

Yasmina Abd‐Elhakim

Academic Editor

PLOS ONE

Journal Requirements:

   "No fund"

    "The researchers would like to thank the Deanship of Scientific Research, Qassim University 

for funding publication of this project."

     "No fund"

Reviewers' comments:

Reviewer's Responses to Questions

**Comments to the Author**

1. Is the manuscript technically sound, and do the data support the conclusions?

Reviewer #1: Yes

Reviewer #2: Yes

Reviewer #3: Yes

2. Has the statistical analysis been performed appropriately and rigorously? 

Reviewer #1: No

Reviewer #2: Yes

Reviewer #3: Yes

3. Have the authors made all data underlying the findings in their manuscript fully available?

Reviewer #1: No

Reviewer #2: Yes

Reviewer #3: Yes

4. Is the manuscript presented in an intelligible fashion and written in standard English?

Reviewer #1: Yes

Reviewer #2: Yes

Reviewer #3: Yes

5. Review Comments to the Author

Reviewer #1: Interesting study entitled "Health status outcome among cannabis addicts after treatment of addiction".

This is a very interesting study as it is the first study to evaluate the impact on health status after treatment of cannabis addiction.

The manuscript is written with attractive and novel results for the scientific community.

If the authors complete the analyzes requested below to strengthen the results, this paper could be a very strong piece of work that could be accepted for publication in PLOS ONE.

However, some elements need to be addressed by the authors to help improve the quality of the manuscript:

MAJOR COMMENTS:

1) In methods, can you provide what treatment patients received, and mention in the discussion and limitations if it could interact with the samples blood test performed in this study.

2) Please provide the number of lost of follow up.

3)Concerning figure and tables:

- consider to add a flowchart

- Please in figure 1, 2, 3, 4, please provide p-value between the different columns. A table with all the data summarized with 4 columns should be considered (Control, Cannabis users, cannabis users after treatment, and p-value)

- In table 5, 6, 7 and 8, authors mentions a correlation between CWAS, cannabis level, age, and duration of cannabis intake, BMI, and vitals signes. It is unclear how variables are corelate. Unless I’m mistaken, Person correlation variable under 0.5-0.7 is not a positive correlation, even if p-value is significant. There is no p-value needed in this context.

- It could be interesting to perform some subgroups analysis (in younger and older patients for example)

3) At the beginning of the discussion, authors usually summarized the majors findings and results of the study. This work evaluates biological and vital signs of an intervention which is the treatment of cannabis addiction. Please, choose the most important results to prove your point.

3) This study about cannabis and his pleiotropic effect on human body is very interesting. But the second part of the study is also assessing the impact of the treatment of the addiction. Do you have a proof of the weaning of the patients by a scale?

4) Authors didn’t address the limitations of the study in the discussion.

5) Authors didn’t discuss about the significant difference of level of oxidative stress biomarker between the 2 groups at the admission and then didn’t normalize after treatment in groupe 2.

6) In the conclusion, authors should be more precise and factual about the findings.

MINOR COMMENTS:

4) In table 1 legend, I didn’t understand why you referred to alcohol group ? Is-it a mistake ?

5) The authors should use a native english speaker to correct minor grammatical mistakes.

8) I think it would be easier to read if the authors rename groupe 1 and 2 with control groupe and cannabis user group.

Reviewer #2: This manuscript describes a study that evaluated changes in health status before and after treatment using body mass index, glycaemic status, liver function, renal function, and oxidative stress in cannabis addicts.

The study was carried out on 100 hospitalized patients receiving addiction treatment and 20 healthy volunteers. Pre- and post-treatment health status indicators were determined by using cannabis level in urine, Body Mass Index, vital signs, some biochemical and oxidative stress parameters. While no change was observed in the BMI values and oxidative stress parameters between the groups before and after the treatment, it was revealed that there was a significant decrease in biochemical parameters in the dependent group after the treatment. Current findings reveal that cannabis addiction damages various body systems and has significant effects on the health status of the addict. This is an interesting finding that could lead to further research, confirming the phenomenon and attempting to identify the cause.

It is seen that the differences in the data obtained especially in the discussion section of the article are not fully revealed. The data obtained while writing the discussion and articles about it were written. However, as the last sentence, the author's opinion on why this result may have occurred has not been added. For this reason, I think that eliminating the deficiencies given below in detail will be important for the quality of the article.

Overall the study appears to have been conducted appropriately and results interpreted correctly (with some exceptions mentioned below). The article reads well, but there are some deficiencies that need to be edited from beginning to end, as there are places that cause misunderstanding at some points. The fact that this study lasted for two years raises fundamental concerns about the rationale for this study. Specific concerns that need to be addressed are as follows:

1. While it was stated that the dependent group was selected from male members, gender was not specified in the control group.

2. It should be scientifically stated why a two-year period was chosen to continue this study.

3. It was stated that the samples were collected both at the time of application and after the treatment. It is a matter of curiosity whether any measures have been taken to prevent this interval (approximately 2 years) from causing differences in the test results.

4. In particular, how to distinguish whether the differences in the results obtained from the evaluation methods used in the material method section over the years are due to the treatment or to normal physiological processes.

5. "Males were more likely than females to abuse substances, and most began using substances in high school." How did you come to this conclusion?

6. "Compared to the control group, current results do not show a significant reduction in BMI at admission and after treatment." Can you explain why?

7. Could you indicate the reason why the vital signs in Group II were significantly lower in the discussion section?

8. It is predicted that the fluctuation in blood glucose level occurs.

9. “Our findings found that in comparison to the control group, there was a significant increase in the mean levels of albumin, total bilirubin, direct bilirubin, AST, ALT, and ALP in group II at admission, but no significant change in group II after treatment”. Please explain why this situation has arisen.

Reviewer #3: It is good study. If not out of scope please add a couple of para on the benefit of THC in reducing inflammation. You can also state a bit about CBD if that is okay. The benefit of this study is far from seen.

6. PLOS authors have the option to publish the peer review history of their article (what does this mean?). If published, this will include your full peer review and any attached files.

Reviewer #1: **Yes: **Antoine Léquipar

Reviewer #2: No

Reviewer #3: **Yes: **Syed Aziz

---

## [Decision Letter · Decision Letter 1]

21 Jul 2023

PONE-D-23-02435R1Health status outcome among cannabis addicts after treatment of addictionPLOS ONE

Dear Dr. Emara,

Thank you for submitting your manuscript to PLOS ONE. After careful consideration, we feel that it has merit but does not fully meet PLOS ONE’s publication criteria as it currently stands. Therefore, we invite you to submit a revised version of the manuscript that addresses the points raised during the review process.

We look forward to receiving your revised manuscript.

Kind regards,

Yasmina Abd‐Elhakim

Academic Editor

PLOS ONE

Journal Requirements:

**Additional Editor Comments:**

It is highly recommended to attach a separate file of response to reviewer comments.

Reviewers' comments:

Reviewer's Responses to Questions

**Comments to the Author**

1. If the authors have adequately addressed your comments raised in a previous round of review and you feel that this manuscript is now acceptable for publication, you may indicate that here to bypass the “Comments to the Author” section, enter your conflict of interest statement in the “Confidential to Editor” section, and submit your "Accept" recommendation.

Reviewer #2: (No Response)

Reviewer #3: All comments have been addressed

2. Is the manuscript technically sound, and do the data support the conclusions?

Reviewer #2: (No Response)

Reviewer #3: Yes

3. Has the statistical analysis been performed appropriately and rigorously? 

Reviewer #2: (No Response)

Reviewer #3: Yes

4. Have the authors made all data underlying the findings in their manuscript fully available?

Reviewer #2: (No Response)

Reviewer #3: Yes

5. Is the manuscript presented in an intelligible fashion and written in standard English?

Reviewer #2: (No Response)

Reviewer #3: Yes

6. Review Comments to the Author

Reviewer #2: Since I could not reach the "author response" content, which includes the answers to the corrections I have written, I could not re-evaluate. for this reason, I need an "author response" that responds to each of my criticisms in a neat and sequential manner. Therefore, I requested a major revision again.

Reviewer #3: The article is better now. The topic is quite relevant and from an area where such studies are not much common so an article like this is good study

7. PLOS authors have the option to publish the peer review history of their article (what does this mean?). If published, this will include your full peer review and any attached files.

Reviewer #2: No

Reviewer #3: No

---

## [Author Response · Author response to Decision Letter 1]

26 Jul 2023

Please the attached files.

PONE-D-23-02435R1

Health status outcome among cannabis addicts after treatment of addiction

PLOS ONE

Dear Dr. Yasmina Abd‐Elhakim

Academic Editor

PLOS ONE

Journal Requirements:

Thank you, I agree

All references are revised and corrected.

Additional Editor Comments:

It is highly recommended to attach a separate file of response to reviewer comments.

Thank you, I agree

Reviewers' comments:

Reviewer's Responses to Questions

Comments to the Author

1. If the authors have adequately addressed your comments raised in a previous round of review and you feel that this manuscript is now acceptable for publication, you may indicate that here to bypass the “Comments to the Author” section, enter your conflict of interest statement in the “Confidential to Editor” section, and submit your "Accept" recommendation.

Reviewer #2: (No Response)

Thank You, 

My reply to your previous round comments:

Reviewer #2: This manuscript describes a study that evaluated changes in health status before and after treatment using body mass index, glycaemic status, liver function, renal function, and oxidative stress in cannabis addicts.

The study was carried out on 100 hospitalized patients receiving addiction treatment and 20 healthy volunteers. Pre- and post-treatment health status indicators were determined by using cannabis level in urine, Body Mass Index, vital signs, some biochemical and oxidative stress parameters. While no change was observed in the BMI values and oxidative stress parameters between the groups before and after the treatment, it was revealed that there was a significant decrease in biochemical parameters in the dependent group after the treatment. Current findings reveal that cannabis addiction damages various body systems and has significant effects on the health status of the addict. This is an interesting finding that could lead to further research, confirming the phenomenon and attempting to identify the cause.

It is seen that the differences in the data obtained especially in the discussion section of the article are not fully revealed. The data obtained while writing the discussion and articles about it were written. However, as the last sentence, the author's opinion on why this result may have occurred has not been added. For this reason, I think that eliminating the deficiencies given below in detail will be important for the quality of the article.

Overall the study appears to have been conducted appropriately and results interpreted correctly (with some exceptions mentioned below). The article reads well, but there are some deficiencies that need to be edited from beginning to end, as there are places that cause misunderstanding at some points. The fact that this study lasted for two years raises fundamental concerns about the rationale for this study. Specific concerns that need to be addressed are as follows:

1. While it was stated that the dependent group was selected from male members, gender was not specified in the control group.

Thank you, I agree 

We did in the methodology section.

2. It should be scientifically stated why a two-year period was chosen to continue this study.

Thank you, I agree 

Two-year period, to be sure of weaning from cannabis use

3. It was stated that the samples were collected both at the time of application and after the treatment. It is a matter of curiosity whether any measures have been taken to prevent this interval (approximately 2 years) from causing differences in the test results.

Thank you, I agree 

The sample withdrawn immediately after finishing the treatment, after that the patient under the follow up period.

4. In particular, how to distinguish whether the differences in the results obtained from the evaluation methods used in the material method section over the years are due to the treatment or to normal physiological processes.

Thank you, I agree 

The sample withdrawn immediately after finishing the treatment.

5. "Males were more likely than females to abuse substances, and most began using substances in high school." How did you come to this conclusion?

Thank you, I agree 

This conclusion was obtained from previous studies and we delete it.

6. "Compared to the control group, current results do not show a significant reduction in BMI at admission and after treatment." Can you explain why?

Thank you, I agree 

Energy uptake, storage, and conservation are primarily mediated by CB1 receptors. Through a number of techniques, it works to maximize energy uptake and conservation. Food is made more palatable by modulating taste and smell pathways by stimulating CB1 receptors [33]. There is a lot of proof that THC or cannabis use causes the CB1R to be downregulated. Regular cannabis usage is linked to CB1R downregulation and desensitization, and CB1R levels stay low for 3–4 weeks after use has stopped [34]. During this time, recently abstinent users would have decreased hunger and elevated metabolic rates. They won't, however, continue to have short-term stimulation of appetite, increased energy consumption, and decreased metabolic rates. Therefore, until CB1R levels reach those from before cannabis usage, weight won't rise because energy intake and storage are still suppressed and metabolism is increased [35]. Ngueta et al., 2015 reported that a higher metabolic rate in marijuana users explain the lack of association between cannabis use and BMI, despite a high energy diet [36]. These data explained the findings of the current study. As compared to the control group, the current results do not indicate a significant reduction in BMI at admission and after treatment.

7. Could you indicate the reason why the vital signs in Group II were significantly lower in the discussion section?

Thank you, I agree 

Immediately following marijuana use, the heart rate and blood pressure may increase [35]. Increased sympathetic nervous system activity is responsible for the increased heart rate, supine systolic and diastolic blood pressures, and forearm blood flow caused by marijuana cigarettes [36]. After a few repeated dosages, tolerance to several of these effects becomes noticeable. After just a day or two of repeated exposure, there may be significant tolerance to changes in blood pressure and elevated heart rate. Tolerance is lost just as quickly when marijuana or THC use is discontinued. Blood pressure falls and the resting heart rate drops after a few days of frequent oral THC exposure or a marijuana plant extract [37].

8. It is predicted that the fluctuation in blood glucose level occurs.

Thank you, I agree 

The present study demonstrated that, compared to the control group, there was a significant increase in the mean fasting blood glucose level values of cannabis user group at admission but no significant change in cannabis user group after treatment. After treatment, the mean fasting blood glucose levels in cannabis user group decreased significantly compared to their respective at admission values. Marijuana use was linked to lower levels of fasting insulin and homeostasis model assessment of insulin resistance [37]. Cannabis use for 12 months is linked to nearly double the risk of diabetic ketoacidosis in one study of adults with T1D when compared to no cannabis use [38]. The variations in cannabis strain potencies may help to explain the variations in blood glucose levels. The potent preparations ('ganja' and 'charas') suppress appetite whereas the weak preparation ('bhang') induces anorexia. THC levels in peripheral blood in people who have smoked one or more marijuana cigarettes have exceeded the range of 5 to 250 ng/ml, whereas intravenous delivery of 10 mg/kg THC produced blood levels of 720 ng/ml after 20 minutes [9]. Other research has demonstrated that cannabis influence the stress response and increase the release of adrenocorticotropin (ACTH) [39, 40].

9. “Our findings found that in comparison to the control group, there was a significant increase in the mean levels of albumin, total bilirubin, direct bilirubin, AST, ALT, and ALP in group II at admission, but no significant change in group II after treatment”. Please explain why this situation has arisen.

Thank you, I agree 

Hepatotoxicity is a possible side effect of using illegal drugs, such as marijuana. A study by Borini et al. evaluating clinical and laboratory changes in chronic marijuana users showed about fifty percent of patients have enlarged spleen or liver or both. Also, marijuana use has been linked to an increase in serum levels of AST, ALT, and AP, regardless of the amount or duration of use [40]. Our findings found that in comparison to the control group, there was a significant increase in the mean levels of albumin, total bilirubin, direct bilirubin, AST, ALT, and ALP in cannabis user group at admission, but no significant change in cannabis user group after treatment. These findings are consistent with a study by Devinsky et al. from 2018 which discovered that liver enzymes increased in chronic cannabis users and returned to normal following the drug's administration [41]. Additionally, Wani et al., 2017 found that cannabis users had lower levels of total protein, albumin, and globulin and higher levels of total bilirubin, bilirubin direct, bilirubin indirect, aspartate amino transferase, alanine amino transferase, alkaline phosphate, and gamma-glutamyl transferase in comparison to the control group [42]. Cannabis causes hepatotoxicity as described by Borini et al. (2004) and Ewinget et al. (2019) due to cholestatic damage, which is supported by an increase in ALP. Cannabis is less likely to cause liver fibrosis, which could account for the remission of liver abnormalities [40, 43].

Reviewer #3: All comments have been addressed.

Thank you

2. Is the manuscript technically sound, and do the data support the conclusions?

Reviewer #2: (No Response)

The current findings show that cannabis addiction harms various body organs except the kidneys and seriously affects addicts' health. This study revealed that after treatment, there was no improvement in oxidative stress biomarker values, but there was an improvement in the other biological parameters. 

Reviewer #3: Yes

Thank you

3. Has the statistical analysis been performed appropriately and rigorously?

Reviewer #2: (No Response)

Data were collected, tabulated, and statistically analyzed using the paired samples t-test to determine whether there were any significant differences between the data of the cannabis user group before and after treatment. Results will be expressed as means ± SD, and statistical significance will be determined if the p-value is less than 0.05. In addition, Pearson correlation coefficients were used to determine the relationships between the CWAS, THC levels, age, and duration of addiction with BMI, vital signs, renal, and hepatic parameters. The SPSS Version 21.0 [SPSS, Inc., Chicago] package will be utilized for all statistical analyses.

Please see the result section for our findings. It is revised and reformulated.

Reviewer #3: Yes

4. Have the authors made all data underlying the findings in their manuscript fully available?

Reviewer #2: (No Response)

All data are present as a separate file named supporting information.

Reviewer #3: Yes

5. Is the manuscript presented in an intelligible fashion and written in standard English?

Reviewer #2: (No Response)

The article was paraphrased and revised (see certificate) 

Reviewer #3: Yes

6. Review Comments to the Author

Reviewer #2: Since I could not reach the "author response" content, which includes the answers to the corrections I have written, I could not re-evaluate. for this reason, I need an "author response" that responds to each of my criticisms in a neat and sequential manner. Therefore, I requested a major revision again.

Dear Thanks for your comment and I am ready to answer any further comment.

Reviewer #3: The article is better now. The topic is quite relevant and from an area where such studies are not much common so an article like this is good study

Thank you

7. PLOS authors have the option to publish the peer review history of their article (what does this mean?). If published, this will include your full peer review and any attached files.

Do you want your identity to be public for this peer review? For information about this choice, including consent withdrawal, please see our Privacy Policy.

Reviewer #2: No

Reviewer #3: No

---

## [Decision Letter · Decision Letter 2]

15 Aug 2023

Health status outcome among cannabis addicts after treatment of addiction

PONE-D-23-02435R2

Dear Dr. Emara,

We’re pleased to inform you that your manuscript has been judged scientifically suitable for publication and will be formally accepted for publication once it meets all outstanding technical requirements.

Kind regards,

Yasmina Abd‐Elhakim

Academic Editor

PLOS ONE

Additional Editor Comments (optional):

**Reviewers' comments:**

**Reviewer #2:** I am of the opinion that the subject of the study is compatible with what is discussed in the journal and at the same time, the arrangement of the article is in accordance with the “Plos-One” guidelines. I am of the opinion that the corresponding author has made the corrections requested by me appropriately, and therefore, it can be published as the original article in its final form.

---

## [Editor Report · Acceptance letter]

25 Aug 2023

PONE-D-23-02435R2 

Health status outcome among cannabis addicts after treatment of addiction 

Dear Dr. Emara:

I'm pleased to inform you that your manuscript has been deemed suitable for publication in PLOS ONE. Congratulations! Your manuscript is now with our production department. 

Kind regards, 

on behalf of

Prof. Dr. Yasmina Abd‐Elhakim 

Academic Editor

PLOS ONE